# Nursing Enlightenment and a Grudge—Reinventing the Medieval Virgin's Benevolent Breasts

Cristina M. Guardiola-Griffiths

Department of Languages, Literatures, and Cultures, University of Delaware, Newark, DE 19716, USA; cmgm@udel.edu

**Abstract:** This article expands upon the function of an adulterous episode in Chapter X, Book II of Alfonso Martínez's *Corbacho*. The tale of adulterous deception may use, parodically, *Madonna lactans* imagery to reveal women's sinful nature, the extent to which may be understood through an expression of inverted forces. That is to say, the import of Marian lactation in artistic and literary representations helps to fully address women's particularly evil ways. A final consideration will be given to a particular image of Marian lactation, which represents the hagiographic legend of St. Bernard of Clairvaux (1090–1153).

**Keywords:** *Madonna lactans*; misogyny; *Corbacho*; breastfeeding; Bernard of Clairvaux





## 1. Introduction

The work of Alfonso de Martínez de Toledo, the *Libro del Arcipreste* (better known as the *Corbacho*), was specifically written to inform of the sins of man. Its author, Alfonso Martínez de Toledo, points to women as a sole source of that sin, even while the outrageous and obscene behavior exhibited by the women of this tractate prompts the reader (or listener) to laughter. One example of such laughable and lustful behavior is found in Book II, chapter ten: "De cómo la muger miente jurando e perjurando" [How a woman lies and perjures when she swears the truth].[1] It is the first and longest of four *enxiemplos* within the *Corbacho* chapter, each of which describe how an adulterous wife eludes punishment for her infidelity. This first story is neatly described as follows: "Wife shows husband how full her breasts are of milk. She squirts milk in his eyes and lets lover escape (K1516.9/T481.1)."[2] The salacious and deceptive use of breasts underscores the Archpriest's arguments exposing women's vice. These arguments are made all the more clear through their implicit juxtaposition with Marian representations of maternal devotion, represented both in the text, in the literature of the Castilian Middle Ages, and in *Madonna lactans* iconography.

Adulterous deceptions, of course, are not the sole purview of Martínez's invective, nor is Martínez's *Libro* their only literary course. These deceptions figure largely in medieval and folkloric literature, and they may be found in indexes compiled by Thompson and Goldberg. In the tales catalogued by these scholars, cheating wives fool their witless husbands and go unpunished for their infidelities. Goldberg has discussed the sexual humor of these tales, and has noted that

> jocular tales . . . are perplexing if we choose to consider them as tales aimed at attacking women. Even a casual reading suggests that the real target is either the complaisant husband, who is so expertly cuckholded, or the naïvely unaware ascetic, who is tempted sexually by the supposedly wicked woman. (Goldberg 1983, p. 69)

Goldberg leans on contemporary theorists and the conviction of fourteenth-century Castilian poet, Juan Ruiz, with the understanding that the "introduction to humor . . . would impart wisdom to his readers" (Goldberg 1983, p. 71). Whether in the *Libro de buen amor* or in collections of *exempla*, these stories serve as moral examples for those who read or

listened to them. In the four stories in the tale, as well as in stories found in French *fabliaux*, Italian *novella*, or other tales of adulterous deception, paramours are hidden by cloth or by darkness, and husbands are tricked through misdirection or by their own gullibility. In most of these examples, something is placed in the way of the husband's sight that interferes with his perception. It is something tangible such as a bedsheet or a frying pan, or intangible such as darkness or a lie. In all cases, truth is obfuscated. In all cases, mundane infidelities between husband and wife are retold in ways that seem to explain women's wickedness. The closest resemblance to the Archpriest's story is the following variation: "The husband's good eye is treated. The wife pretends to treat his one good eye, so that he cannot see the paramour" (K1516.1/T481.12). This summary describes tales told in the *Libro de los enxemplos*, the *Esopete historiado*, and in the *Disciplina clericalis*. These cases of infidelity give the author many examples upon which to base his tale.

What sets the archpriest's advisement apart from the others lies, in part, on the implicitly hostile exchange between husband and wife. In the Archpriest's story, the wife effectively blinds her husband's eyes with her breastmilk. He reacts with displeasure and pain: "¡O fija de puta, cómo me escuece la leche!" (Gerli 1992, p. 188) [Oh, you hussy, how the milk stings!].[3] The verb, *escocer*, appears twice in the *Corpus del Nuevo Diccionario Histórico del Español* before making its appearance in the *Corbacho*. The first is a reference to the properties of the radish; the second, circa 1411, comes from an analogy in one of St. Vincent of Ferrer's sermons, which claims that falling prey to vice is like picking a scabrous itch: one is relieved during the picking, but the sting reappears and is accompanied by regret. The breast milk's sting can be similarly read. The husband's temporary blindness may be understood as a commentary on the quality of the wife's milk, but it may also be a critique on the quality of her morals. Studies on wetnurses in the Middle Ages put forth the idea that the quality of a woman's breastmilk was related to her virtue; the shot to the eyes may then be seen as a commentary on his wife's loose ways.[4] It bears mention that this woman is not only a wife, but also a mother. Her tryst informs how we read her, her behavior, and its outcome. It allows us to see how her actions may be subversive within a patriarchal community. That the wife is also a mother of a young child suggests, logically, that the child may be illegitimate. The moral objective of this medieval treatise implies something far more than the evil of a wife's adulterous lie. The *desordenado amor* provoked by this woman condemns the lover's soul, subverts the husband's authority, and calls into question the legitimacy of her offspring. If this example was meant to show how the Archpriest viewed all women, then his claim that women are the root of all evil would be understandable. Martínez concludes by insisting that the chapter must be read without ulterior motive:

> E aunque seré de algunos reprehendido por non saber ellos mi entinçión–la qual solo Dios sabe en este paso non ser a mala parte–porque algunas cosas pongo en prática dirán que más es avisar en mal que corregir en bien. Diga cada qual su voluntad, que yo non lo digo por que lo así fagan, mas porque sepan que por mucho que ellos nin ellas encobierto lo fagan e fazen, que se sabe, e algunos sabiéndolo, a sus mugeres, fija e parientas castigarán. (Gerli 1992, p. 190)

> And some people will scold me (since they don't understand my intention, which in this instance God only knows is not wicked) because I show how some things are done, saying that I'm more interested in giving lessons in how to put wickedness into practice than in giving good examples to follow. Let everyone say what he thinks best. But I'm not relating this so that they'll do things that way but so they they'll know that, no matter how much they (either men or women) do or might do in secret, that their tricks are known and that some men, because they know it, will be able to give proper lessons to their wives, daughters, and female relatives. (Naylor and Rank 2011, p. 134)

The chapter should not be misunderstood, and so the Archpriest makes his intentions clear: the chapter should not enable the wicked to sin.[5] It is as if the Archpriest was fearful that his message would be lost in the interplay between the text and its context. The

Archpriest explains that his writing needs to be used to reprimand the wicked with "proper lessons" to improve behavior.

The *Corbacho* story is one of many medieval exempla on adulterous deception that appear in various works, times, and contexts. But it is the only one involving breastmilk. The singularity of this breastmilk deception makes it likely that the Archpriest chose it specifically. Lactation is a biological phenomenon associated with motherhood; the act of breastfeeding is often associated with idealized nurturing and maternal care. For example, lactation is central to several miracle tales extolling the virtues of the Virgin Mary. Stories about Mary the Mother of God tell how her milk was used to heal the penitent, restore faith, and reveal Christian truth. Within the context of our Archpriest's story, the oppositional nature of human and divine mother's milk establishes an ironic doubling. The uniqueness of the detail in the Archpriest's sinful "Eve" stands in contrast to the presence and sacred power of Mary—and her breasts. In this Ave/Eva dichotomy, the Archpriest's "Eve" underscores the problematic nature of female sexuality while the implied Mary elides this nature through her immaculate maternity. References to Mary appear throughout the four books that make up Archpriest's book. She appears repeatedly as an advocate for humankind, as a humble virgin, and as the mother of humankind's redeemer. But at the same time, the Archpriest insists on women's evil character. This figural inversion of Mary and Eve forms the underpinning of studies such as James Burke's *Desire Against the Law*, wherein the natural correspondences between what is sacred and barbarous provides a way to fully explore meaning in works in medieval literature. Ryan Giles uses this idea as a foundation for explaining saintly parodies in his *Laughter of the Saints*. This author explains how a saint's image may be understood through an exchange between oral and written traditions, which focus (although perhaps not exclusively) on the saint's name, life, attributes, and characteristics of the community in which he or she is revered. Giles furthers this understanding by noting that medieval texts contain registers, which point to different communities of learning. The learned and clerical elite might have renegotiated meaning "through ecclesiastical practices, texts, and institutions" (Giles 2009, p. 12), but also through popular registers. This is the case for works such as the *Libro de buen amor*, in which popular verse forms, vernacular language, as well as "proverbs, extra-liturgical rituals, legends, and superstitions" are mixed with classical and ecclesiastic authority to provoke humor (Giles 2009, p. 12). This imitative interplay between popular and learned is evident in many works from the late medieval and early modern periods, and the *Corbacho* is no exception. Michael Gerli explores the style and structure of the Archpriest's work, which resembles a frank and familiar preaching style used by the Dominican order (Gerli 1969, p. 107). This affective use of dialogue, Gerli adds, can overwhelm the work's purpose, resulting in characters whose language seems to take control of the work. Their language, still today, produces humor that confuses the moral message.

A similar example may be found in the sermons of preachers such as Bernardino de Sienna [1390–1444]. Bernardino, whipped up by the zealous frenzy of his oratory, occasionally over-powered his religious message when addressing popular concerns.

> "O, o, del latte della Vergine Maria; o donne, dove siete voi? E anco voi, valenti uomini, vedestene mai? Sapete che si va mostrando per reliquie: non v'aviate fede, ché elli non è vero: elli se ne truova in tanti luoghi! Tenete che non è vero. Forse che ella fu una vacca la Vergine Maria, che ella avesse lassato il latte suo, come si lassa delle bestie, che si lassano mugnare? (Mormando 1999, p. 283)

> And, oh, oh, by the way, the milk of the Virgin Mary! Ladies, where are your heads? And you, fine sirs, have you seen any of it? You know, they're passing it off as a relic. It's all over the place. Don't you believe in it for a moment. It's not real. Don't you believe it! Do you think the Virgin Mary was a cow, that she would give away her milk in this way—just like an animal that lets itself be milked? (qtd. in Rubin 2009, p. 300)

The rough language used by the Sienese Franciscan to describe the Mother of God is far from reverent. Equating Mary with a cow might produce a shameful laugh, overwhelming the Christian goals of the sermon, which served to reinforce Mary's ideal qualities. By likening her through a negative example to a cow, Bernardino offers a rational argument for casting doubt upon the many reliquaries containing her breastmilk. Bernardino's words attest to a widespread image of a nurturing, lactating Mother of God, even while his suspicion of the holy milk reliquaries casts doubt on the validity of these religious objects. Bernardino's speech allows for the degradation of these objects, even while the source of these items remains pure and intact.

This juxtaposition of contrary objects may form a way of explaining or understanding what Burke (1998) calls the "middle mode" of understanding; through the presentation of things scandalous, a deeper understanding of the moral or doctrinal message could be realized (5–6). This view explains the complementary nature of an adulterous wife here representing, parodically, a lactating Mary. While the former is a mother of a newborn child of questionable parentage, the latter is the Mother of God, whose virgin birth has offered humankind the possibility of Christian redemption. The extent of the wife's sinfulness is revealed through Mary's idealized behavior. It is reasonable to suggest that these contrasting images were used by Martínez as a means to convey his book's misogynist objectives. The remainder of this article will address the importance of Marian lactation in artistic and literary representations, furthering the idea that Marian imagery could have been easily understood as a counterpoint to the folkloric motif of the second book. The widespread use of this image and the development of the artistic representation of the *Madonna lactans*, especially the representations that include the hagiographic legend of St. Bernard of Clairvaux, reveal the truth of Mary's redemptive role for humanity at the same time that its expression in the Archpriest's story exposes the sinfulness of female nature.

## 2. The Virgin Mother of God: Mary's Cultural Importance

Images, statues, and stories from the ancient Mediterranean address the importance of maternity through representations of birth and lactation. In stories of ancient Egypt, statues of the divine Egyptian goddess Isis show her nursing her son Horus. Early Greek pottery shows the suckling demigod Heracles, whose bite caused Zeus's wife, Hera, to produce the Milky Way.[6] Lauren Rodríguez Peinado (2013) notes the incorporation of these images and stories into Christian doctrine, which eventually allowed for an analogous female mode of divinity. While the Virgin Mary was not always considered divine, her rise in standing within the Church may have begun with the circulated stories collected in early Christian Apocrypha (especially the Protoevangelium of St. James). These stories told about her life, representing her in ways that will be further developed in the centuries that follow. She is a chaste and holy child, a docile vehicle for God's will, and a nurturing young mother. Miri Rubin notes that between the first and fifth centuries of the Common Era, Mary emerged from her more modest role in the Gospels to hold a more important role as Bearer of God (Rubin 2009, p. 88). In subsequent centuries, early Christian bishops occasionally wrote about the Mary's virginal nature and her role within the Church. Of particular interest is Ildefonsus, archbishop of Toledo (607–667), whose treatise *De Virginitate Sanctae Mariae* (*On the Perpetual Virginity of Saint Mary*) was considered by Marcelino Menéndez y Pelayo to be the "*primer monumento literario exclusivamente consagrado entre nosotros a la devoción de Nuestra Señora*" (Menéndez y Pelayo 2012, p. 225) [The first literary monument exclusively consecrated among us to the devotion of Our Lady]. The life of this peninsular bishop would be told again and again, indirectly propagating Marian ideology. Julio Vélez-Sainz lists the Latin literary diffusion of this Iberian saint. *A Vita Beati Ildefonsi Archiepiscopi Toletani* within the *De viri ilustribis* (690), a *Vita vel gesta S. Ildefonsi Toletanae Sedis Metropolitani Episcopi* (ca. 774–783), Rodrigo Manuel el Cerratense's *Vita Beati Ildefonsi, Archiepiscopi* (thirteenth century), and the *Legenda Beati Illefonsi, Archiepiscopi Toletani secundm regulam Asturicensis Ecclesiae* (late thirteenth century) are the predominant works in Latin to have been written about Ildephonsus in the Middle Ages. In Castilian, there are also the *Istoria*

*de Sant Alifonso, Arçobispo de Toledo* and the *Vida de San Alifonso por metros* (Vélez Sainz 2008, p. 142). In addition, three vernacular versions of Ildephonsus's miracles appear in the royal codices of Alfonso X the Wise.[7] Ildephonsus's life also appears in the first miracle of Berceo's *Milagros de nuestra señora*. Most importantly, the Archpriest of Toledo, Alfonso Martínez, compiled a hagiography for Ildephonsus and translated the treatise that won him saintly fame.[8]

Mary's role within the Church slowly expanded. In addition to the biblical writings and hagiographic tales, Christian liturgical calendars also included days devoted to the Virgin Mary. Mary was observed throughout the year, especially in relation to Christ's life. Rubin specifies that the Mozarab liturgical calendar included no fewer than six days throughout the year during which Mary served as a focal point or an indirect object of religious reflection (Rubin 2009, p. 94).[9] All these feast days either directly or indirectly deal with the contradictory nature of maternity and virginity, which are also elements taken up in the artistic representations of Mary. Quoting from Groen, Rubin takes the words of a seventh-century pope:

> When I enter a church, I contemplate images of Jesus Christ's miracles and his mother suckling Our Lord, and Our Lord in her arms, while angels around them sing a hymn *sanctus, sanctus, sanctus* (Rubin 2009, p. 98)

Mary's ideological development between the fifth and fifteenth centuries responds to two concerns within Christian dogma. The first focuses on her status as Mother of God (*Theotokos*), which was promoted at the Council of Ephesus (431); the second focused on the conditions regarding Mary's incarnation, conditions that were still being debated in the fifteenth century. Both developments, in a sense, addressed Mary's worthiness as a maternal vessel for God. By the close of the Middle Ages, Christian liturgy within the Iberian peninsula promoted Mary's maternal importance by emphasizing her purity, beauty, and unblemished nature (Twomey 2008, p. 22).

Mary's idealized, maternal role for humanity, fostered within medieval Christian doctrine, correlates to the rise of literary representations of the Virgin Mother. While these representations may have originated from apocryphal gospels or tales representing the Holy Family's flight into Egypt, it is safe to add that the cult of the Virgin also grew from writings of later theologians and spiritual writers, who would have used Eve and others as figures meant to reveal Mary's finer qualities.[10] Certainly, the writings of monastic houses and the preaching of mendicant orders in the latter centuries of the Middle Ages promoted the role and importance of the Virgin Mary within Christian communities of faith. The earliest collection of Marian miracle tales compiled by both Cistercians and Dominicans included lactational miracles (Warner 1976, p. 198).

A maternal Mary was instrumental in the development of dogma that espoused man's salvation through the Virgin's intercession.[11] Within the Iberian peninsula, Gonzalo de Berceo developed Marian theology through his devotional miracle poems. Mary's motherhood is central to understanding Berceo; his writings showcase Mary's relationship to Christ in a way that explains Christian dogma. Berceo's inclusion of Marian miracle "*La casulla de San Ildefonso*" ("Saint Ildephonsus's Vestment) indirectly affirms the aforementioned seventh-century Archbishop's writings on Mary's virginity (and, therefore, her aptness for her role as Mother of God).[12] As Gregory Andrachuk explains, the use of the word "*adonado*" was rare for its day; he suggests that Berceo's usage was a deliberate choice that explains the divine favor bestowed upon Mary (Andrachuk 2017, p. 537).[13] Motherhood is an essential figuration of the Virgin Mary, and her ability to heal through nurturing sustenance a natural derivation of that figure. In Berceo's Marian miracle "*El clérigo y la flor*" (The Cleric and the Flower"), Mary identifies herself by her maternal qualities:

> Demandóli el clérigo que yazié dormitado,
>
> "¿Quí eres tú que fablas? Dime de ti mandado,
>
> ca cuando lo dissiero seráme demandado

quí es el querelloso o quí el soterrado."

Díssoli la Gloriosa: "Yo so Sancta María

madre de Jesu Christo que mamó leche mía;

el que vos desechastes de vuestra compannía,

por cancellario mío yo a éssi tenía. ([Gerli 1988](#), p. 90)

The cleric, who had been sleeping, asked Her:/"Who are you who speaks? Tell me, whom you command/for when I say this I will be asked/who the aggrieved one is or who the buried one is."/The Glorious One responded: "I am Holy Mary,/Mother of Jesus Christ, who suckled My milk./The one you excluded from your company,/I held as a chancellor of Mine." ([Grant Cash and Mount 1997](#), p. 37)

Mary justifies her command that the cleric bury the sinner on hallowed ground because she is the mother of the Christ child. Her authority rests on the very maternal power of nursing. This identifying feature of Mary's lactational power may also be seen in the *Cantigas de Santa Maria*. These songs were written in Galician-Portuguese during the reign of Alfonso the Learned (1221–1284) and are often attributed to the wise Castilian king. Cantiga 422 clearly explains the importance of mother's milk for humanity. The song "*Madre de Déus, óra pro nós téu Fill'essa hóra*" [Mother of God, pray to your son on our behalf at that hour] prefigures the wrath of God on Judgment Day. In it, Mary is beseeched repeatedly to intervene and save man from eternal damnation.

E u el a todos parecerá mui sannudo

entôn fas-ll' enmente de como foi concebudo....

E en aquel día, quand' ele for mais irado,

fais-lle tu emente com' en ti foi enserrado....

U verás dos santos as compannas espantadas,

móstra-ll' as tas tetas santas que houv' el mamadas....

E u mostrar ele tod' estes grandes pavores,

fas com' avogada, ten vóz de nós pecadores

que polos téus rógos nos lév' ao paraíso

séu, u alegría hajamos por sempr' e riso. ([Casson 2022](#), stanzas 2–4, 19–20)

And when he appears to all in great wrath,/then make him remember how he was conceived . . . ./And on that day when he is most wrathful,/make him remember how he was enveloped by you . . . /When you see the frightened hosts of saints,/show him your holy breasts which he sucked . . . /And when he reveals all these terrible things,/take on the role of Advocate and plead for us sinners,/so that, because of your prayers, He take us to His paradise/where we may have joy and laughter forevermore. ([Kulp-Hill 2000](#), pp. 508–9)

In addition to this prayer-like Cantiga, no fewer than five other songs touch upon Mary's breasts as a source of miracle and salvation.[14] These are *cantigas* 46, 54, 138 and 404 and may be found summarized in the [Oxford Cantigas de Santa María Database](#) ([2005](#)):

Cantiga 46: The Moor who Venerated an Image of the Virgin Mary

Cantiga 54: The Monk who was Healed by the Virgin's Milk

Cantiga 93: The Leper who was Healed by the Virgin's Milk

Cantiga 138: John Chrysostom's Vision

Cantiga 404: The Priest who was Healed by the Virgin's Milk

(Oxford Cantigas de Santa María Database, accessed on 21 March 2022)[15]

In Cantiga 138, Mary's breastmilk is noted for its nourishment and the pleasure the baby Jesus derives from it (Vaz Leão 2007, p. 122). While John Chrysostom does not receive the milk, it is his awareness of its power to feed and to please that leads to the miraculous recuperation of his eyesight. The milk serves as a poetic motif connecting human to divine sustenance, projecting upon the former the redemptive powers of the latter. While this instance does not reveal breastmilk as instrumental to the production of the miracle, other *cantigas* identify the breastmilk as the catalyst that promotes spiritual (faith) or physical healing. The Muslim man in Cantiga 46 is at first unable to accept Christianity, since he cannot accept the dogma of a God made man. The Muslim, moreover, is incredulous that God would have accepted being born of a woman: "*que non podia creer/que Deus quisess'encarnar/nen tomar/carn' en moller*" (vv. 38–41) [Because he could not believe/that God would wish to become incarnate/nor been born of a woman]. The two disbeliefs are intertwined, since the disbelief of Mary as *Theotokos* predicates the Moor's initial rejection of the Christian faith. It is, therefore, telling that acceptance of Jesus's divinity is proved by Mary's ability to nurse. As the statue's breasts flow with milk, the Moor is converted. Mary's maternal projections, those life-giving mounds of nutrition, are again the key to divine salvation.

The three remaining *cantigas* more specifically highlight Marian mammary miracles. In Cantigas 54, a devout Cistercian (*monge branco*) lies dying of a throat ulcer. Mary cleanses the wound and then she " . . . *deitou-lle na boca e na cara/do seu leite*" (vv. 60–61) [poured milk upon his mouth and face], after which the monk was healed. In Cantiga 93, a burgher's son repents from a life of debauchery through prayer. The son's daily recitation of 1000 Hail Marys motivates for the Virgin's pity. The *cantiga* uses the verb "anoint" "*seu santo leite o corpo li'ongiu*" [her holy milk anointed his body] (v. 38) to describe how the Virgin delivers the son from his physical affliction. Cantiga 404 has, perhaps, the most questionable of miracles. Like Berceo's fornicating monk, the monk of Cantiga 404 deserves Mary's aid only because of his constant worship. While our modern sensibilities disdain the oral and performative mechanics of ritual prayer, privileging an association of interior thought with sincere devotion, outward, public manifestations of devotion were the norm at least until the close of the Middle Ages. This outward devotion is what saves the monk; Mary comes to his rescue, anointing his mouth with healing, miraculous breast milk.

It is safe to say that Mary's nurturing, maternal role for humanity, fostered within medieval Christian doctrine and amplified by literary narrative, was readily understood by medieval Christian laypersons. This identification of Mary as a nurturing mother was also furthered in the artistic productions of the late Middle Ages. Cecelia Dorger has argued that both Dominicans and Franciscans "increased their efforts to teach visual meditative practices to the laity that, in effect, resulted in spiritual communion" (Dorger 2012, p. 128). Dorger's thesis, in part, focuses on the combined impact of the mass and *Madonna lactans* altarpiece paintings, which allowed for a coalescing of physical and spiritual communion for the churchgoer. Dorger's thesis focuses on Italian images of the *Virgo lactans*, which were introduced in eastern parts of the Iberian peninsula and painted in the International Gothic style during the latter half of the fourteenth century. Icons of this Virgin depict her seated on a cushion and in the intimate act of breastfeeding the Christ child. Nuria Blaya Estrada notes that these paintings of the Virgin gave the viewer a means to understand their religious significance . . .

> y esto hace que las pinturas, no solamente narrativas, sino también las puramente rituales o devocionales, se hagan ahora más humanas, más realistas, más accesibles a la devoción personal". (Blaya Estrada 1995, p. 163)

> And this allows painting, not merely the narrative ones, but also those purely ritualistic and devotional, become more humane, more realistic, more accessible to personal devotion.

Blaya Estrada points out that the comfortable seating position and intimate act of the nursing *Madonna* connected the faithful Christian to the divine through personal analogy

(Blaya Estrada 1995, p. 163). The Virgin Mary is on a cushion and not a throne; her dress is simple and not adorned in regal splendor. She is more like a commoner than a queen. This unassuming, human depiction of Mary is the focal point of the artistic representation. Even while a panoply of angels or stars depict Mary's holiness, her domesticity (all the more apparent in her act of nursing) broke with strict medieval hierarchies representing Mary as divine Mother of God. This made her relatable, more comparable to all mothers.

There is some debate as to the earliest Iberian production of the *Madonna lactans* painting, but it is certainly true that this style of painting was in the Iberian peninsula during the Archpriest's life. The earliest production of the *Madonna lactans* painting is attributed either to Ramon Destorrents or Jaume Serra. The Virgen del Tobed is a panel painting currently housed at the Prado Museum in Madrid (Figure 1). It can be dated with relative certainty to between 1359 and 1362 due to the identity of the patron and his family. Enrique II of Castile (1378–1404), his wife and two children appear in the lower corners of the painting.[16] At least four different artists had workshops that produced versions of the lactating Madonna. The aforementioned artists, Jaume Serra (who belonged to a family of artists working during the fourteenth and fifteenth centuries) and Ramon Destorrent, were presumably eclipsed in quality by Ramón de Mur and Lorenzo Zaragoza. This last painter was also known as the Master of Villahermosa. Pere IV of Aragón (1319–1387) was known to have described Zaragoza as "lo millor pintor que en aquest çiutat sia" [perhaps the best painter in this city] (Arciniega García 1995, p. 32; Miquel Juan 2015, p. 503). These four artists were active during the latter half of the fourteenth and early fifteenth centuries, and at least five different tempera panels of the *Madonna lactans* can be attributed to these artists and their workshops.[17]

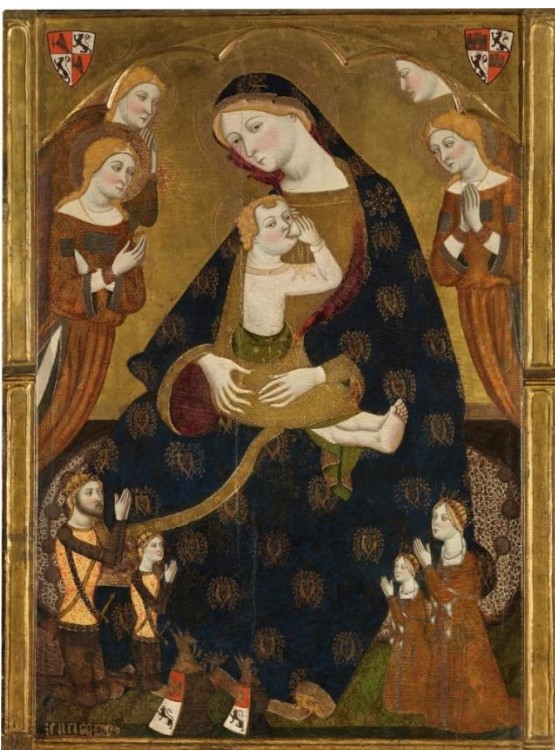

**Figure 1.** Jaume Serra. The Virgin of Tobed with the Donors Henry II of Castile, his Wife Juana Manuel, and two of their Children, Juan and Juana. (Lembrí, Pere Museo del Prado 2022) © Museo Nacional del Prado.

Given the spiritual possibilities afforded by the *Madonna lactans*, through which analogies with Christ's life and with the Church's objective of spiritual redemption may be found, it is unsurprising to see churches consecrated to the Virgin Mary. It goes without saying

that the more churches were consecrated to the Virgin Mary, the more the associations of Mary (and her redemptive, maternal qualities) are present in that community's life. Some of these churches predate the above-mentioned panel paintings, such as the carved wood statue of the *Virgen del Rebollet*, which represents a nursing Madonna and is found in Oliva (Valencia), and dates from the twelfth or thirteenth centuries. A polychromatic granite statue of the *Virgen de Oseira,* also a lactating Madonna, dates from the thirteenth century, and may be found the eponymous monastery in Cea (Galicia). In the shrine of Our Lady of Miravalles (Asturias), the *Virgen de Miravelles* is a carved and painted granite stature of a nursing Madonna dating somewhere between the eleventh and thirteenth centuries. A *Madonna lactans* image appears on a frieze in the Iglesia de Tarrega (Lérida) circa 1269, in a relief adorning the Puerta de Platerías in the Cathedral of Santiago, and as a miniature on parchment belonging to Jaume I of Aragón (1213–1276).[18] All of these artistic works attest to the growing popularity of the Virgin Mary, whose role within the Church and in the culture of the late Middle Ages was still evolving. The numerous examples of this religious icon, and the particular use of the panel painting, both in the Iberian peninsula and throughout the continent, confirm the prevalence of the Virgin as a visual devotional object.

Images of a nursing Mary adorned many churches and monasteries in the Iberian peninsula, but they were also images actively and contemporaneously being produced during the lifetime of the Archpriest of Toledo. The pervasiveness of *Madonna lactans* imagery is further demonstrated through the work of Giovanni Dominici (1355–1419). In his *Regola del governo di cura familiare*, an instruction manual on the care and management of the family, he advises using the following religious images for the education of children:

> La prima si è d'avere dipinture in casa di santi fanciulli o vergine giovanette, nelle quali il tuo figliuolo, ancor nelle fascie, si diletti come simile e dal simile rapito, con atti e segni grati alla infanzia. E come dico di pinture, cosi dico di scolture. Bene sta la Vergine Maria col fanciullo in braccio, e l'uccellino o la melagrana in pugno. Sarà buona figura Iesu che poppa, Iesu che dorme in grambo della Madre; Iesu le sta cortese innanzi, Iesu profila ed essa Madre tal profilo cuce. (Dominici 1860, p. 131)

> The first rule is to have pictures of saintly children or young virgins in the home, in which your children, still in swaddling clothes, may take delight and thereby be gladdened by acts and signs pleasing to childhood. And what I say of pictures applies also to statues. It is well to have a Virgin Mary with the Child in arms, with a little bird or apple in His hand. There should be a good representation of Jesus nursing, sleeping in His Mother's lap or standing courteously before Her while they look at each other. (Dominici 1927, p. 34)

Dominici calls for an educational role of the *Madonna lactans* image, and in doing so he brings the icon into the home. It is hard to avoid seeing now the ready connection between Mary and all mothers. The Mother of God functions as a role model for all medieval women; the nurturing, nursing qualities of the Virgin have extended into the home to idealize what a mother should be and how a mother should act. The pervasiveness of the *Madonna lactans* image is apparent throughout Christian society, extended to a populace through religious sermon, Church reliquary, and domestic iconography. It stands to reason that this image of the *Madonna lactans*, in either word or picture, could easily have informed the story described by Alfonso Martínez in Book II, chapter ten of the *Corbacho*, and served as the implicit counterpoint to the negative example of the adulterous young wife and mother.

## 3. A Strange Hypothesis: The Lactation of St. Bernard

The previous section explored the prevalence of *Madonna lactans* imagery within Christian communities, thereby suggesting that the image would have been readily understood as a counterpoint to the negative medieval example narrated in the Archpriest's second

Book. The variety and quality of *Madonna lactans* tales and icons notwithstanding, no one tale or icon shares as many parallels with the Archpriest's story as the Lactation of St. Bernard. The story was first putatively told in Conrad of Eberbach's *Exordium magnum* (d. 1221).[19] This thirteenth-century tale of Cistercian miracles includes the life of St. Bernard of Clairvaux and may recount a legend in which St. Bernard prayed for revelation before a statue of the Virgin and Child. The statue responded to that prayer by coming to life and nourishing the Cistercian monk with her milk. Variations of this legend describe different locations of anointment. Mary's milk falling onto Bernard's mouth has been interpreted as the miraculous catalyst enabling the monk to speak the true word of God. Other legends depict Mary's milk falling upon Bernard's eyes, allowing him to see Mary's divine nature. In either of these cases, breastmilk reveals divine truth and allows for the spreading of God's Word.

The St. Bernard narrative must have circulated during the late Middle Ages in the Iberian peninsula, even if no extant hagiographic tale of the saint may be found that dates from the Archpriest's life. St. Bernard's influence in these geographic realms may be seen with a cursory view of Rafael Durán's *Iconografía española de San Bernardo*. Durán includes 100 plates of more than 80 images, 27 of which are from the late Middle Ages. Most of these were painted or sculpted in the cities and towns of the Crown of Aragón (e.g., Tarragona, Mallorca, Barcelona, and Castellón), and many refer to hagiographic episodes of the monk's life. Discoveries are still being made. As recently as 2004, Manuel Sánchez Mariana brought to light the existence of a manuscript containing Bernard of Clairvaux's sermons; these he dated to the first quarter of the thirteenth century and located to the Leonese monastery of Santa María de Sandoval (Sánchez Mariana 2004, pp. 1361–74). St. Bernard appears as the interlocutor with the Virgin in the *El duelo que fizo la Virgen María el día de la pasión de su fijo Jesucristo*.[20] It is certainly curious that the monk from Cantiga 54 in Alfonso the Learned's Marian miracles tales is identified as a white, or Cistercian, monk. It is equally curious that hagiographic narratives relate Bernard's frequent illnesses, again like the monk in the 54th cantiga. A fifteenth-century Book of Hours, the *Horas de Philippes Bigota*, housed in Madrid's Biblioteca Nacional (BNM RES/281), shows an illumination of the Cistercian St. Bernard. Although he is not accompanied by a statue of the Virgin Mary, the text that wraps around the illuminated letter is taken from Psalm 12: "*Illumina oculos meos, ne umquam obdormiam in morte; ne quando dicat inimicus meus, praevalui adversus eum*" (Libro de Horas 1490–1500, fol. 180r) [Enlighten my eyes that I never sleep in death: lest at any time my enemy say, "I have prevailed against him"]. These words highlight Bernard's sight, representationally important for the miracle that distinguishes him. A fifteenth-century *santoral* presumed to have been owned by the Counts of Haro also contains the life of Saint Bernard. Although devoid of the Marian miracle, the importance of breastmilk is paramount to the story. In the hagiographic tale, Bernard's mother Alech (or Alicia)

> ... pario siete fijos & los seys varones & la vna fembra & los varones fueron monjes & la fenbra monja Et luego que pario el fijo lo tomaua en las manos & lo ofrescia a jhesu christo & criaualos a sus tetas dandoles con la leche las costumbres que auia ella mesma e cresciendo en quanto estauan sso su poderio & rregimiento mas los criaua para monjes que para seglares (Leyenda Aurea 1400–1499, fol. 35r)

> ... gave birth to seven children, six sons and one daughter. The sons were all monks and the daughter a nun. After [giving] birth to a child she would take him in her hands and offer him to Jesus Christ. And she suckled them at her breast, transferring her virtues through her milk. And while they were raised under her authority and tutelage, she raised them to become monks rather than laypersons.

Here, Bernard's mother functions in a manner similar to the statue from Eberbach's tale, gifting the child the nourishment of Christian virtue through the intimate process of breastfeeding.

While the lactational legend of St. Bernard has not surfaced in fifteenth-century Iberian narratives, there are numerous artistic representations of this legend.[21] The earliest

appearance is a Mallorcan altarpiece dedicated to St. Bernard, dated circa 1290, and attributed to the Master of Palma (Figure 2). The altarpiece is said to have been part of a reconsecrated oratory within a conquered Muslim fortress called the Almudaina de Gumara (Salvadó 2006, p. 50). It belonged to the Knights Templars, for whom the Cistercian monk was instrumental.[22] The altar is divided into five panels: the central panel houses a painting of the saint, while four flanking panels recount hagiographic episodes of his life. The scene found on the upper left post of the altarpiece depicts the lactation vision granted to St. Bernard.

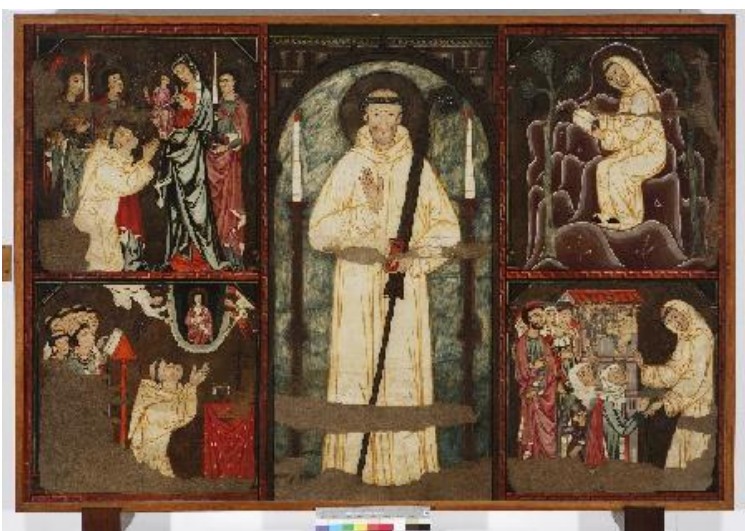

**Figure 2.** Retaule de Sant Bernat de Claraval. © Museu de Mallorca.

The altar panels' French style may suggest the legend originated north of the Pyrenees; the style bears no resemblance to the Italian paintings of the *Madonna lactans* (Bauer 2015, p. 79). Durán notes two other retable representations of the *lactatio* miracle from the 1300s; the first, circa 1348, from the church of Santa María de Montblanch (Tarragona) and the second, probably from the monastery of Benifassá (Castellón) (Durán 1990, pp. 43–46; plates XII–XIV). The first is a polychrome stone retable dedicated to Saints Bernard and Barnabus and is attributed to Guillermo Timor. The central relief displays the two saints. Both sides of the central relief are divided into four equal quadrants, which include episodes from each saint's life. Even though it is badly damaged, the bottom left quadrant to the left of the central relief still displays the *lactactio*. The last retable is catalogued by the Museo del Prado as *La Virgen de la Leche con el Niño entre san Bernardo de Claraval y san Benito* (Figure 3).

While it is thought to come from the Cistercian monastery of Nuestra Señora de Benifazá in Castellón, it was later moved to the Iglesia del Priorato de Santa Ana de Mosqueruela in Teruel.[23] Mary appears as the central figure in a position similar to the *Madonna lactans* panel paintings earlier discussed. Angels flank either side of her. St. Bernard appears with an open book on Mary's right and receives Mary's lactational nourishment. St. Benedict is placed on Mary's left and holds another open book. Durán suggests that these three *lactatio* images provide the origins of the Bernardine tradition. Salvadó echoes Durán when he notes that, other that the Palma Master altarpiece, "[n]o other surviving Spanish artworks from the late 12th to 13th centuries depict St. Bernard" (Durán 1990, p. 52). Salvadó also notes for the Crown of Aragón that Saint Bernard paintings were found only in Cistercian monasteries during the fourteenth century (Salvadó 2006, p. 52).

By the early fifteenth century, there is one retable that can be traced to a convent not associated with the Cistercians. The Sivera Retable (ca. 1415) comes from the Santo Domingo de Valencia convent. The Bellas Artes Museum in Valencia, where this retable now is housed, confirms that the work was created to adorn the Dominican monastery, and further identifies the artist to be Antoni Peris, who was active in Valencia between 1404 and

1423 (Figure 4). While the retable is incomplete, the central panel still shows a *Madonna lactans* figure. The right *calle* divides into three sections; the uppermost depicts the Bernard *lactatio*.

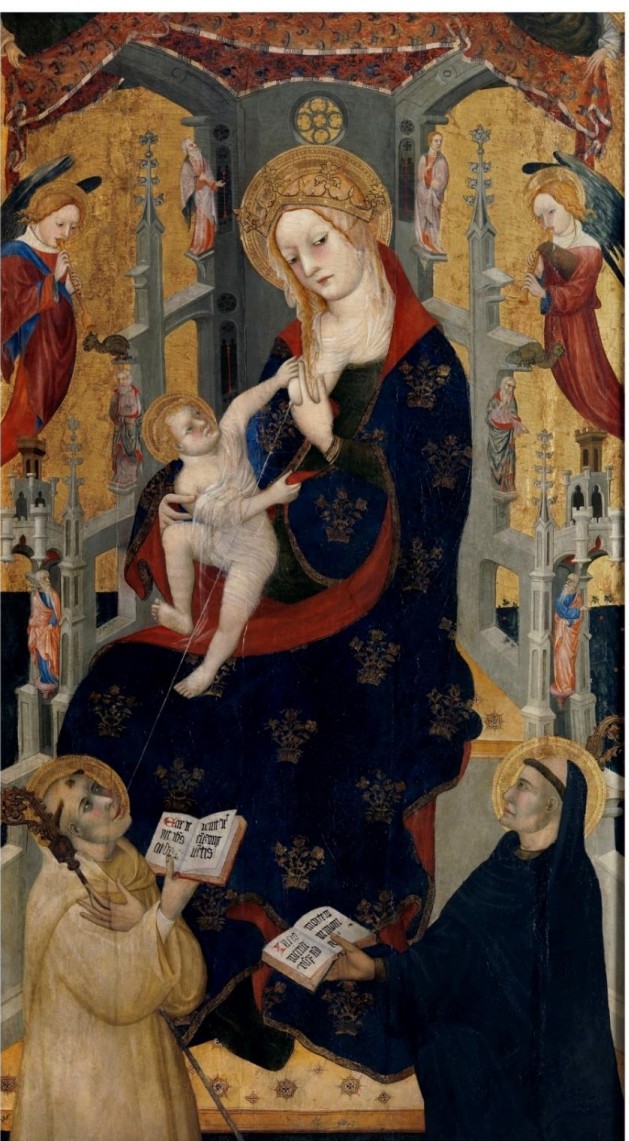

**Figure 3.** Pere Lembrí. *The Virgin nursing the Child between Saints Bernard of Clairvaux and Benedict.* ©Museo Nacional del Prado Virgen del Tobed, Museo del Prado.

The stone retable from Santa María de Montblanch, the Sivera Retable and the Prado Museum's *Virgen de la Leche con el Niño* are suggestive of the incorporation of the Bernard miracle into the *Madonna lactans* iconography. By the first quarter of the fifteenth century, one may see this diffusion within various churches, monasteries, and convents in the Crown of Aragón, where it was possible for Alfonso Martínez to have come upon it. Alfonso Martínez lived in several cities in the Crown of Aragón between 1420 and 1430, while in search of greater benefices from the Church. Naylor and Rank note Valencia, Tortosa, Barcelona, and Gerona as places that the archpriest may have visited (Naylor and Rank 2011, p. 3). Many of these towns and cities were centers of development for *Madonna lactans* paintings. The Sivera Retable is perhaps the most likely of candidates. Although Alfonso Martínez did not make profession to the Dominicans, his association with the Order may be seen in his preaching style or in the relationship with his Dominican patron, the Cardinal San Sixtus. These are indicators of a relationship that may have afforded him a stay within the Valencian monastery where the retable was located.

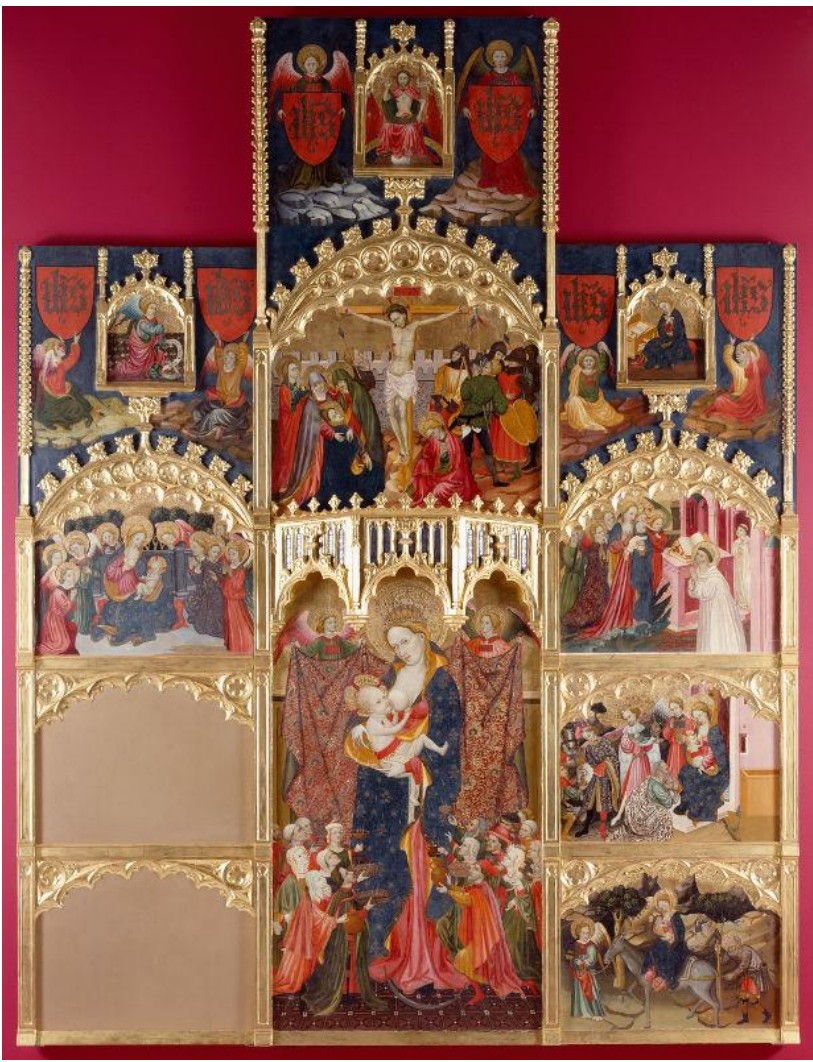

**Figure 4.** Antoni Peris. *Retaule de la Mare de Déu de la Llet*. © Museu de Belles Arts de València.

Whether or not the Sivera Retable or St. Bernard's lactational miracle lies at the heart of the *Corbacho* story cannot be conclusively proven. But it is certainly true that Saint Bernard is well represented in Iberian art of the late fourteenth and early fifteenth centuries. It is equally true that Bernard's life and lactational miracle do not disappear from the record.[24] In the *Cancionero de* Úbeda (1588), following the romance dedicated to Saint Ildephonsus, Don Justo de Sancha includes another dedicated to Saint Bernard. The first twelve verses are as follows:

> Por su virtud y limpieza
>
> El melifluo san Bernardo,
>
> Por su devoción tan alta
>
> La Virgen un don le ha dado.
>
> Hizole su capellán
>
> Muy querido y regalado,
>
> Y estando ante él un día
>
> En oración transportado,
>
> Puso la Virgen la mano
>
> En su pecho consagrado,

Y con divina leche

Los labios le ha rociado. (Biblioteca de Autores Españoles 1855, ll. 1–12; p. 120)

The Virgin has given the mellifluous Saint Bernard a gift on account of his virtue and integrity and his supreme devotion. She made him her beloved and prized chaplain. While he was transported through prayer one day, appearing before him the Virgin placed her hand on her consecrated breast and moistened his lips with divine milk.

Bernard's breastmilk miracle must have been so well known that by 1605, the Spanish *Vida de san Bernardo* alludes to it as commonplace knowledge:

Todo esto fue efecto de aquel tan supremo regalo con que la Sacrasantissima Virgen y sin mancilla Maria endulço su voz, su pluma, y su lengua, siendo para todos atractivo suaue su doctrina. Y aunque es dulcissimo en todas sus Obras, Y Escritos este Docto Melifluo; pero donde incomparablemente vierten suauidad sus Sermones y Platicas, es en los Mysterios Inefables, y Sacrosantos del Hijo de Dios, como son, su Encarnación y Glorioso Nacimiento; y en las alabanças de la Sanctissima Virgen Maria, todo es suauidad, agudeza, y dulçura. Mamólo todo de los Pechos desta Reyna Virgen, Y Madre, para que suauemente atraxesse al seruicio de su Hijo, y suyo a los hombres: y desterrasse con su predicacion las amarguras, y azedias de la culpa, con lo sazonado, y meloso de sus palabras. (Almonacid 1682, pp. 446–47)

All of this was the effect of the supreme gift with which the most sacred and pure Virgin Mary sweetened his [Bernard's] voice, his pen, and his tongue, making his teaching tender and attractive. Even though this mellifluous doctor is very sweet in all his works and writing, where his Sermons and Letters pour the most tenderness is in the discussion of ineffable and sacrosanct mysteries of the Son of God, which are, his incarnation and glorious birth. Additionally, in the praises of the most holy Virgin Mary, he is all tenderness, insight[fulness], and sweetness. He suckled this all at the breasts of our Virgin Queen and Mother, so that he might gently be drawn to the service of her Son, and so that men be drawn to his service, and so that he might banish with his preaching all sorrow and remorse over blame, with his sweetly expressive words.

Returning to the Archpriest's story, one may see inverted structural elements between Saint Bernard's story and that of the cuckholded husband. The wife and husband reflect, as would an inverted mirror, the relationship between Virgin and Saint Bernard. The Virgin's breastmilk nourishes this saintly figure who reveals God's truth. In artistic representations of the *Madonna lactans*, the Virgin's milk provides nourishment for the Son of God. As the maternal advocate for all of humanity, her milk is spiritual nourishment for all of humankind. As the inversion of this spiritual food, the wife's breastmilk stings or burns. The wife uses it not to promote well-being but to keep her husband from seeing. The Archpriest's story can be read as a figural example revealing the woes of human vice. The adulterous wife, unchaste, un-nurturing, un-Christian, holds no redeeming qualities. Unlike the Virgin Mary, she advocates sinfulness instead of salvation. The adulterous wife pours breastmilk onto her husband's eyes not to "sweeten" his sight but to effectively blind him. Unlike Bernard, the husband cannot see the truth of his wife's infidelity.

One can reach similar interpretative revelations in all the men represented in the lactational stories identified in this article. The men in all these stories, be they sinners, cuckold, or Saint Bernard, are all living their lives, well or poorly, according to Christian norms. Even the Moor, whose deeds overseas suggest his involvement in the Crusades, takes for himself a statue of the Virgin, and so suggests his openness to conversion. By straight or twisted means, all these stories suggest a model of Christian life. To return to Burke's understanding of contrasting images—not as binary, oppositional images, but as images meant to fully reveal an idea through explorations of its inverse—one may see

the gravity of the sin of adultery through the redemptive virtue of Mary's motherhood. The Virgin Mary, and particularly her breastmilk, allows for sinners in the *Cantigas* to understand and accept the truth, the Christian way of life. Symbolically, the milk serves as a form of nourishment for man, an idea attested to in both Old and New Testament (Isaiah 60:16: 1 Peter 2:1–3). For the Cistercian monk in the *Cantigas*, Mary heals his throat and mouth, the parts of his body needed to proclaim the Good Word. In the many representations of the *Madonna lactans*, we see that what nurtured the baby Jesus in physical form serves to nourish all men spiritually. Through Saint Bernard's *lactatio*, we see the import of material intercession for divine revelation.

It is plausible to assume that Martínez's story parodically used Bernard's legend as its backdrop. In both tales, the lactating mother allows her breastmilk to fall upon the eyes of a man seeking truth. In the case of the adulterous wife, a physical truth is denied; for the Cistercian monk, spiritual truth is revealed. The sacrosanct, healing nature of Mary's milk is understood more fully through the noxious sting of the wife who betrays her husband. The Archpriest comically emphasizes the sinful nature of woman in order to expose the frailties of men's souls in the work he has finished "*aviendo por medianera, intercesora, e abogada a la humill sin manzilla virgen Sancta Maria*" (Gerli 1992, p. 61) [having as intermediary, intercessor, and advocate the humble and immaculate Virgin Holy Mary] (Naylor and Rank 2011, p. 25).

**Funding:** This research received no external funding.

**Institutional Review Board Statement:** Not applicable.

**Informed Consent Statement:** Not applicable.

**Data Availability Statement:** Not applicable.

**Conflicts of Interest:** The author declares no conflict of interest.

## Notes

[1] Michael Gerli's edition of the *Arcipreste de Talavera o Corbacho* has been used in this article. English translations are based on Eric Naylor and Jerry Rank's translation, albeit in some cases slightly modified. All other translations, unless noted, are mine.

[2] Tales of adulterous deception play largely in the catalogue of folklore motifs identified by Stith Thompson (1955–1958) and Harriet Goldberg (1998), but this one is unique to the Archpriest's story. The complete episode is as follows: "Contarte he un enxiemplo, e mill te contaría: una muger tenía un ombre en su casa, e sobrevino su marido e óvole de esconder tras la cortina. E quando el marido entró dixo: "¿Qué fazes, muger?" Respondió: "Marido, siéntome enojada." E asentóse el marido en el banco delante la cama, e dixo: "Dame a çenar." E el otro que estava escondido, non podía nin osava salir. E fizo la muger que entrava tras la cortina a sacar los manteles, e dixo al ombre: "Quando yo los pechos pusiere a mi marido delante, sal, amigo, e vete." E así lo fizo. Dixo: "Marido, non sabes cómo se ha finchado mi teta, e ravio con la mucha leche." Dixo: "Muestra, veamos." Sacó la teta e diole un rayo de leche por los ojos que lo cegó del todo, e en tanto el otro salió. E dixo: "¡O fija de puta, cómo me escuece la leche!" Respondió el otro que se iva: "¿Qué debe fazer el cuerno?" E el marido, como que sintió ruido al pasar e como non veía, dixo: "¿Quién pasó agora por aquí? Paresçióme que ombre sentí." Dixo ella: "El gato, cuitada, es que me lieva la carne." E dio a correr tras el otro que salía, faziendo ruido que iva tras el gato, e çerró bien su puerta e tornóse, corrió e falló su marido, que ya bien veía, mas non el duelo que tenía" (Gerli 1992, p. 188). [I'll tell you a tale, and I could tell you a thousand. A woman had a man in her house and her husband came back unexpectedly, and she had to hide him behind the curtain. And when her husband came in he asked: "What are you up to, wife?" She answered: "Husband, I feel a little indisposed." And her husband sat down on the bench in front of the bed and said: "Give me some supper." And the other man who was hidden [neither could] nor dared to leave. And the woman pretended that she was going behind the curtain to get out the tablecloths and said to the man: "When I stick my tits in front of my husband, scram, sweetheart, and get yourself out of here." And this is what she did. She said: "Husband, you don't realize how my breast has swelled; and I'm dying because there's so much milk." He said: "Let's see: show me." And she took out her breast and gave him a squirt of milk in the eye and completely blinded him, and in the meanwhile the other man started out. And the husband said, "O son of a bitch, how the milk smarts." The other man who was going out answered: "I bet horns hurt more!" And the husband, since he heard noise as the other man passed by and since he couldn't see, said: "Who just passed by? I thought I heard a man." She said: "It's the cat, woe is me, and he's carrying my meat off." And she started running after the other man who was leaving, making noise like she was going after the cat, and she shut the door tight and went back and found her husband who could see fine now but who couldn't manage to see his own grief" (Naylor and Rank 2011, p. 133).

[3] This translation does not follow Naylor and Rank in order to preserve the gendered insult.

4   See Emilie Bergmann (2002) and Caroline Castiglione (2013) for the cultural significance and social practices regarding breastfeeding, and the use of wetnurses. Much of our understanding about breasfeeding in the Middle Ages comes from studies that point out how wetnurses were chosen and for how long they were employed. Curiously, one of the first literary mentions to wetnurses exemplifies the before mentioned need to select the proper woman. The *Libro de Alexandre* notes that the ancient Macedionian king "nunca quiso mamar leche de mugier rafez,/si non fue de linaje o de grant gentilez" (7cd) [never wanted to suckle the milk of a common woman, only a noble or a woman of good manners] (Casas Rigall 2014, p. 6).

5   It is not unusual to see in medieval Castilian narratives an insisting that one be understood correctly. In the Archpriest of Talavera, Juan Ruiz cautions against the deliberate mishandling of his work. Sin is an intrinsic part of human nature. The negative examples in the *Libro de buen amor* are not intended to promote sin, even though those who want to sin will find in the reading a way to do it.

6   An ambivalent connection between mother and suckling child may be seen in *The Libation-Bearers.* In Aeschylus's tragedy, Clytemnestra futilely begs her son Orestes for mercy. Clytemnestra leans on the bond established between mother and suckling child. It is not enough.

7   Anthony Cárdenas notes these versions (chapter 510 of the *Estoria de España*, *cantiga* 2 from the *Cantigas de Nuestra Señora*, and a marginal gloss in mss. T.1.1), although only two are considered to belong to the wise king (Cárdenas 1983, pp. 339–40). These versions refer to one or both of the Virgin Mary's miraculous appearances to the sainted bishop. In the first, she appears with his book in her hands, gratified by his literary defense of her virginity before the heretics who questioned it. In the second, she appears with a fine vestment, a chasuble, also as recompense for his defense.

8   José Madoz y Moleres notes few adaptations in the Archpriest's translation of Ildephonsus's life from the sources used by the fifteenth-century writer. Nevertheless, in the Archpriest's translation of *filius hominis* for hijo de la Virgen, one might understand Martínez's veneration for the Mother of God.

9   These are the celebration of the Nativity (December 25), the Purification (February 2), the Incarnation of the Word in Mary (March 21), the Assumption (August 15), the Nativity of Mary (September 8), and the Annunciation of Mary's Birth (December 18).

10  Lesley Twomey notes that early examples of literature from the Iberian peninsula were marked by devotion to the Virgin and her appearance within the *Protoevangelicum.* Art historians have leaned on this and other apocryphal gospels, as well as on the saintly tales that brought out much of the humanity represented in the figures of the Virgin.

11  Geographical and familial ties lie at the heart of many hagiographic narratives. Saints are often associated with a particular place and are configured as a symbolic father or mother for that village, town, or city. The work of Jacobus da Voragine (1230–1298) expanded the importance of saints beyond the local, tribal level. His *Golden Legend* included the lives of many saints and martyrs; it also included stories that told the life of Mary. The *Golden Legend* extended Mary's profile throughout the European continent and created numerous opportunities to see Mary in the role of divine mother.

12  Martínez de Toledo and de Talavera (2010) composes the hagiographic tale *Vida de San Ildefonso*, in which he emphasizes the virginal purity of Mary. Martínez disabuses those who believed "un omne sabidor . . . Elbidio," whose teachings proclaimed that Mary had not remained a virgin after Jesus's birth. He refutes this idea with a direct appeal to the Virgin on behalf of all humankind, claiming that the righteous know that Mary was sanctified in her mother's womb by the Holy Spirit, and that after having conceived through the Holy Spirit she had a virgin birth of Jesus Christ (Madoz y Moleres 1962, p. 86).

13  Andrachuk's article deals specifically with the definition of the word "*adonado*." For him, the word cannot simply be translated as "full of grace" but rather "full of the gift that is Jesus Christ".

14  Angela Vaz Leão attributes the many references to breastmilk within the *Cantigas de Santa María,* a collection of over four hundred Marian miracle tales, to fecund pagan divinities. She notes, "*[a] força e a extensão do culto mariano se explicam, entre outras causas, pela sua associação ao mito femenino, presente em religiões pré-históricas e atestado por estatuetas de mulhiere nuas, muitas vezes estilizadas e com seus órgãos sexuais exagerados, tais como se entraram em várias escavações, sobretudo no Oriente Médio. Eram imagens da Grande-Mãe, cujo culto a identificava às vezes com a Terra e que assim se espalhou pelo Mediterrâneo*" (Vaz Leão 2007, p. 118). [The strength and wide-spread circulation of the Marian cult may be explained, among other reasons, by her association with the feminine myth, present in prehistoric religions and represented by small statues of female nudes, many of them stylized and with exaggerated sexual organs, as have been found in multiple excavations, especially in the Middle East. These were images of the "Great Mother," whose cult sometimes identified her with the Earth Mother and in this form spread throughout the Mediterranean.].

15  These are the summarized titles given in the Oxford Database. In Portuguese, they are as follows: Cantiga 46: "Esta é como a omagen de Santa Maria, que un mouro guardava en sa casa onrradamente, deitou leite das tetas." 54: "Esta é como Santa Maria guaryu con seu leite o monge doente que cuidavan que era morto." Cantiga 93: "Como Santa Maria guareceu un fillo dun burges que era gafo." Cantiga 138: "Como San Joan Boca-d'Ouro, porque loava a Santa Maria, tiraron-ll'os [ollos] e foi esterrado e deitado de patriarcado; e depois fez-lle santa Maria aver ollos, e cobrou per ella sa dinidade." Cantiga 404: "Como Santa Maria, com seu leite, cura um jovem clérigo seu devoto de grande enfermidade".

16  Enrique II and his family appear in the bottom corners of the tempera panel. Enrique had been in exile in Aragón and under the protection of the Aragonese king, Pere IV. Pilar Silva Maroto notes that Enrique II commissioned three retables for the Nuestra

[17] Señora del Tobed sanctuary in Zaragoza between 1356 and 1359. Notarial documents also indicate that by 1359, the altar for the side panels to the Virgen del Tobed had been completed (Silva Maroto 2006).

[17] The Prado Museum documents Jaume Serra to have been active circa 1358–1390. Destorrents is thought to have worked between 1351 and 1362. Lorenzo Zaragoza may have outlasted both, as he is documented from 1363 until 1406 (Archiniega García 32). Ramón de Mur was active between 1412 and 1435, according to the Museo National d'Art de Catalunya (Mur 2022). There is a curious website (Art+E 2022) with a catalog of *Madonna lactans* images: www.historia-del-arte-erotico.com/cleopatra/home.htm. While offering very little description of the images it contains, this website catalogs a vast number of images throughout the European continent, many of which are painted in the same International Gothic style as those described in this article.

[18] Images of a nursing Eve may also be found within Christian art, which are used as the basis of a figural exegesis foretelling Mary's future redemptive role. Examples of the nursing Eve image can be seen on the eleventh-century Bernward Doors, made for the Hildesheim Cathedral; in the thirteenth-century relief of a portal to the Upper Chapel in Sainte-Chapelle, Paris, and in the stained glass paneling of the Rheims Cathedral of Notre Dame.

[19] Rubin attests to this in her *Mary, Mother of God* (Rubin 2009, p. 150), while Rafael Durán denies it (Durán 1990, p. 40). Others note that the story first appears in art, as part of an altarpiece from a Mallorcan Templar Church, circa 1290. This altar was dedicated to the Cistertian monk and tells his life in pictures (Bauer, Salvadó). Louis Rèau believes that the first images of St. Bernard's lactational miracle may be dated to 1200; the first narrative stories of this miracle may be dated to about 1350 (qtd. in González Zymla 2019, p. 212; Réau 1955–1959, pp. 215–20).

[20] Anthony Lappin (2008) has argued against the attribution of this work to Gonzalo de Berceo, suggesting that only four works can be decisively attributed to this Riojan poet. See the well-received *Gonzalo de Berceo: The Poet and His Verses*.

[21] The Osma Master paints a retable dedicated to Saint Ildephonsus in the Catedral de Burgo de Osma during the second half of the fifteenth century, as does Martín Bernat for a retable in the Tarrazona Cathedral (Zaragoza). An additional altarpiece, attributed to the Master of La Seu de Urgel, whose *Retablo de la Mare de Déu de Canapost* is also known as the *Retablo de la Virgen de la Leche*, dates to the last quarter of the fifteenth century.

[22] Bernard supported the creation of the Order of the Templars during the 1129 Council of Troyes. While there is no consensus as to the part he played in creating the Templar rule, he did write *De laude novae militiae* [*In Praise of the New Knighthood*] to exhort Templar knights in the crusade.

[23] Rafael Durán also includes within his *Iconografía española* a plate from a Cordoban Cathedral Chapter Office of the Virgin with Saints Bernard and Ildephonsus. While the painting contains the *Madonna lactans* icon, the lactational miracle is not present.

[24] Durán traces the appearance of this miracle both in and outside of the Iberian peninsula. He cites its appearance in Guillermo Eysengreinio's *Crónica de Espira* from 1561. Durán also notes the lactational miracle in an *authenticum* dated 1599 from Edmundo de la Croix, the Abbot General of the Cistercian Order to Fr. Fermín Ignacio de Ibero, the Fietro monastery abbot, which attests to the location of the miracle in Châtillon-sur-Seine. A larger narrative of Saint Bernard presumably appears in the *Historia de la esclarecida Vida y Milagros del Bienaventurado Padre y Mellifluo Doctor San Bernardo*, which was a compilation by F. Cristóbal González de Perales that was published in 1601. Cistercian historians make the story more well known in the seventeenth century; their devotion to the saint may be seen in monasteries throughout the eighteenth century (Durán 1990, pp. 41–43).

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
