# Peer review of "Nursing Enlightenment and a Grudge—Reinventing the Medieval Virgin’s Benevolent Breasts"

_religions, doi:10.3390/rel13040326_

Round 1

Reviewer 1 Report

Acept in  present form.

Author Response

thank you for your previous comments.

Reviewer 2 Report

Minor errors:

line 120 - it is not clear to what text you refer when you mention "Mary`s redemptive book"

line 248 - explain the word "adonado"

line 474 - a segment of critics dispute Berceo´s authorship of El Duelo (see for example, Lappin). Needs a footnote to this effect.

line 533 - need to insert the word "book"

Author Response

Thank you very much for your comments. I sincerely appreciate your efforts to improve the quality of my paper.

I am not sure how to reply to the second comment regarding the definition of the word "adonado." Andrachuk's article deals specifically with the definition; for him, the word cannot simply be translated as "full of grace" but rather "full of the gift that is Jesus Christ". I agree with this translation, and I imply the translation when I write that "[Andrachuk] suggests that Berceo’s usage was a deliberate choice that explains the divine favor bestowed upon Mary (537). " 

I did add a footnote, though, to make the definition clear.

I also changed the wording that attributes the "Duelo" to Berceo, and note Lappin's well-received 2018 book that discredits the attribution.

Thank you again.

Reviewer 3 Report

The revisions are careful and complete, renewed compliments to the author.  The documentation is frankly impressive, and so is the critical judgment throughout.

I like the new title and all the careful adjustments.  I apologize for misspelling “Andrachuck” in my earlier review.

My only stylistic suggestion is that several of the paragraphs are too long to make for comfortable reading.  They could be divided into two or three paragraphs that look less imposing.

Add to Keywords ‘breastfeeding’, ‘Bernard of Clairvaux’

Footnote 6: correct spelling “Clytemnestra”

Typo in Bibliography:

Bauer, Doron. “Milk as Templar Apologetics in the St. Bernard of Clairvaux Altarpiece from Majorca.” Studies in Iconography 36 (2015): 79-98.

Author Response

Dear MS evaluator,

Many, many thanks for the careful readings of my article. I am grateful for the help in making this work as best as it could possibly be.

This manuscript is a resubmission of an earlier submission. The following is a list of the peer review reports and author responses from that submission.

Round 1

Reviewer 1 Report

-Some items of the final list of bibliography are only ornamental. This study is not apparently based on all these references, because the author doesn’t quote all of them in the paper (for example, Arciniega, Cáseda, Fogg, Sam and Velasco González, Alberto, etc.). 

-English translations from the Spanish texts should be review by a native English speaker and expert in medieval literature. Sometimes the terms used are anachronistic and unspecific.

- The author should refer precisely to the edition of the Arcipreste de Talavera quoted throughout the paper (Gerli) in the first footnote.

- I highly recommend checking the quoted fragments with the accurate critical edition of the text by Marcella Ciceri, Madrid, Espasa-Calpe, 1990.

-The author should always quote in Spanish the titles of original works first, and then the English translations in brackets, with capital letters in each word, as in English titles. For example, lines 240-241. “La casulla de San Ildefonso” (“Saint Ildephonsu’s Vestment”). And lines 247.  “El Clérigo y la flor” (“The Clerk and the Flower”). These translations are wrong: “Saint Idephonse’s Chasuble” and “The Cleric and the Flower".

- In English translations of Spanish poetry one should mark the end of each verse with slashes. For example, in l. 249: The cleric, who had been sleeping asked Her / “Who are....

-In page 2, line 59, the translation of “fija de puta”, “bitch”, is anachronistic and no literal. Maybe could be better “hussy”.

-The author could connect the words of the prologue in page 3, lines 77-82 with the LBA, 117-132 (ed. Blecua): “Enpero, porque es umanal cosa el pecar, si algunos, lo que non los consejo, quisieren usar del loco amor, aquí fallarán algunas maneras para ello. (...) E Dios sabe que la mi intençión no fue de lo fazer por dar manera de pecar ni por maldezir...”

-What Gerli calls “monologo que parece diálogo” is actually a “dialogismo”. page 4, line. 126.

- On the Perpetual Virginity of Saint Mary. What is the original title? Italics. page. 5, line. 181.

- Footnote 8. The text “tú del ángel anunçiada y yaziendo en el vientre...” must be precisely quoted, at least referring to the precise chapter. As it is explained in Cervantes Virtual this edition is “Edición digital basada en la de Vidas de San Ildefonso y San Isidoro, Madrid, Espasa-Calpe, 1962, pp. 5-64.” It should be recommended checking the analogical edition, because the texts in Cervantes virtual not are always accurate.

-It is striking that a few details are given of the Cantigas de Santa Maria’s manuscripts but  none about the manuscripts and editions of the Arcipreste de Talavera, which is the object of study of this paper (page, 7, line, 267).

- In the Libro de Alexandre, is found an important reference to breast milk: “El infante Alexandre, luego en su niñez, / empeçó a mostrar que serié de grant prez: / nunca quiso mamar leche de mugier rafez, / si non fue de linaje o de grant gentilez (cuaderna 7, ed. Juan Casas Rigall, RAE). The author should read and maybe quote the commentary of this fragment by Ángel Gómez Moreno in Claves hagiográficas en la literatura española (del Cantar de Mio Cid a Cervantes), Madrid/Frankfurt am Main, Iberoamericana/Vervuert, 2008.

-Throughout the article the author mentions some similarities with ancient Greek culture, such as in lines 170-172, but he/she doesn’t refer to an  important scene of the Aeschylus’ Choēphóroi (The Libation-Bearers), where Clytaemestra begs her son, who is about to kill her, for mercy, reminding him that he sucked the milk of her breasts. v. 887: «ΚΛΥΤΑΙΜΗΣΤΡΑ ἐπίσχες, ὦ παῖ, τόνδε δ᾿ αἴδεσαι, τέκνον, / μαστόν, πρὸς ᾧ σὺ πολλὰ δὴ βρίζων ἅμα / οὔλοισιν ἐξήμελξας εὐτραφὲς γάλα.», «Clytaemestra [baring one breast] Stop, my son, and have respect, my child, /  for this breast, at which you many times drowsed / while sucking the nourishing milk with your gums!» (Aeschylus Volume II- Loeb Classical Library, 146. Edited and translated by Alan H. Sommerstein). This scene is also represented in Greek pottery: <https://www.theoi.com/Gallery/T40.9.html>. Museum Collection The J. Paul Getty Museum, Malibu. Catalogue No. Malibu 80.AE.155.1. Beazley Archive No. N/A Ware Paestan Red Figure. Shape Amphora, Neck. Painter Attributed as close to Asteas. Date ca. 350 - 320 B.C. Period Late Classical. This scene could be compared, possibly without citing a source’s real connexion, but at least with a thematic connexion to the Virgin who is repeatedly beseeched to intervene and save man from eternal damnation in “Madre de Déus óra pro nós téu Fill’essa hóra” (see línes 273-282, Filll or Fill?).  

- Original Galician Portuguese titles of Cantigas de Santa María are not quoted in this paper, in lines 293-341.

-The captions of pictures should be written in italics not with quotation marks, such as in figure 1 (lines 413-414), Figure 2 (line 523), Figure 3 (lines 536-537) and Figure 4 (line 558). University of Oxford Style Guide, cfr:        https://www.ox.ac.uk/sites/files/oxford/media_wysiwyg/University%20of%20Oxford%20Style%20Guide.pdf: “Titles of books, journals, plays, films, musical works etc should be given in italics if they are a complete published work; if you are referring to an individual short story, song, article etc within a larger publication, use single quotation marks (see also Quotation marks under Punctuation).”

Reviewer 2 Report

Overall, the article is more a history of the iconography of and literary references to the madonna lactans. While this is interesting, the ostensible focus is the story of adultery from the Corbacho. This is lost in the investigation of the lactating Virgin to the expense of the work by Martínez de Toledo. Also there are numerous grammatical errors, misspellings, and errors, such as in the title of Marina Warner´s well-known book.

The central argument about the use of breast milk in a story about adultery and deception is completely lost in an exhaustive history of the image of the maddonna lactans in Iberia. The author needs to make central a theoretical framework based on the work of James Burke. He/she alludes to Burke´s framework of contrasting images a couple of times, but this should be the axis for his/her arguments. The search for a source for the story in the Corbacho in iconography is not far-fetched but he/she would be better served to carefully investigate other Iberian tales about deceptive, cheating spouses who use "blinding" as a technique to conceal their infidelities.

Reviewer 3 Report

On the Uses of Breastmilk. Parody in the Arcipreste de Talavera

Reader’s Report & Evaluation, June 25, 2021

This is a smart and well-informed study.  Compliments to the researcher.

The only complaint is that despite the promise in the title, we lose sight of the Archpriest’s proposed parody for long stretches of the essay.  I would suggest simply reworking the title so that expectations are not frustrated, perhaps something like “Nursing Enlightenment and a Grudge: Reinterpreting the Medieval Virgin’s Benevolent Breasts”.

The essay also does not need to be quite this long.  There are passages which could be abbreviated like the transcriptions of the Oxford Cantigas.  I know, digital publication had made length irrelevant but that’s led to badly padded articles that no one actually reads all the way through.

Note 2:  The author’s translation in is basically quite good.  In the opening episode I might suggest “frustrated” or “vexed” for enojada rather than “a little indisposed” just to hint at the sexual nature of her extramarital adventurism.  Rather than “When I stick my tits [‘teats’ is the standard international spelling, maybe better for an international readership and for readers many years from now] in front of my husband” perhaps “When I pull out my breasts in front of my husband”: pechos has no salacious tone in Spanish, it’s just a reasonably neutral body part.  For “the other man started out” one might say “the other man bolted out”.  For “he’s carrying my meat off” it may read better as “he’s carrying off my meat”.  That’s a naughty bit for sure, an intruder stealing the household’s poorly guarded flesh.

Lines 69, 157, 218, 242, 623: “thus” is considered archaic and fusty; it can simply be deleted everywhere with any other adjustment.  The word only has appropriate rhetorical force at the end of a sentence when making fun of your own pomposity, thus.

Line 84:  One might put instead of “(… which in this instance God only knows is not wicked)”, “(… which in this instance God alone knows is not wicked)”.  I think the Archpriest’s protests are delivered with a wink.

Line 86: Typo here: “Let everyone say what he things best.”  “Let everyone say what he thinks best.” 

Line 96-97: More compact to say, “The Corbacho story is one of many medieval exempla on adulterous deception in various works, times and contexts.”

Line 110: “Ryan Giles” – full name on first appearance for all authors

Line 135: You are right to quote from Vicente Ferrer and Bernardino de Sienna and others but please include dates for each historical figure as you do for “Jacobus da Voragine (1230-1298)” [with complete numerals].  Both of these authors predate the Corbacho enough to have been available to Martínez and his readership, something to keep in plain view.  The reliance on Bernard of Clairvaux reaches back into the 12th cent., entirely permissible given his enduring popularity, but the temporal distance should be noted as late medieval lay spirituality had evolved considerably away from 12th-cent. monastic spirituality.

Line 151: “But, by likening her through negative example…”   It’s better style never to start a sentence with a single word and a comma.  You can always kill that word and start with the next and let parataxis generate the contrast: “By likening her through negative example…”.

Line 155: “Upon reflection, Bernardino’s speech…”  Same problem: kill that opening phrase.  Also who’s reflecting here?  Bernardino? his listener? the modern reader?  Don’t forget that these sermons are not transcriptions of actual delivery, which may have been even more flamboyant and colorful, but survive as written compositions.  These in turn many have been texts for other friars to preach from or to be read aloud among a small gathering.  Silent reading is still not the norm in the mid-fifteenth cent., especially for texts meant to emulate theatricality as sermons were.  The point stands that sermons retained in published form or even in manuscript are written compositions with different reader responses than those listening to an oral performance.

Lines 162-166:  necessary given the article’s abstract and other preceding material?

Line 168: is there any real difference between ‘import’ and ‘importance’?

Lines 175, 220: where do these apocryphal tales appear?  Can you name specific texts?

Line177: ‘Gospels’ usually with initial capital for the Christian scriptures; lower case suggest metaphorical use, such as “The naïve students took everything their professor said as gospel.”

Note 3: simpler to say, “before the heretics who questioned it.”

Note 4: “in the Archpriest’s translation of filius hominis for “hijo de la Virgen”, one might understand Martínez’s reverence for the Mother of God.”  ‘Adoration of’ is only for God, “veneration/reverence for’ the Virgin and saints.

Line 206: “of a seventh-century pope”: needs a hyphen when centuries are used as adjectives

Line 208: “hymn” instead of ‘hum’?

Line 226: “renounced” or “announced/defended”?

Line 231: “The work of Jacobus da Voragine…”

Line 241-242: “seventh-century | Archbishop’s” needs hyphen

Line 243: “Gregory Andranchuk” full name on first appearance

Line 247: not ‘clerk’ but ‘cleric’

Line 260: “my milk” no capitals

Line 261: “Annette Grant Cash and Richard Terry Mount” full names

Line 265: “Cantigas de Santa Maria” – italics for the title of the compilation, no accent on “Maria” because it’s Galician, not Castilian

Line 269: “Madrew”? check spelling

Line 271: referring to the day of universal judgement or simply to the death of the individual petitioner, as in the common prayer “Hail, Mary”?

Note 8: “virginal purity of Mary”?

Line 363: cantiga in italics

Line 408: “perhaps the best painter in this city” to account for the sia in subjunctive

Line 419: perhaps identification of provinces with parentheses or just a comma rather than square brackets

Line 420: “twelfth or thirteenth centuries” consistently as either words or numerals throughout the article

Lines 426, 478, 547, 563, 564, 572: Aragón should have its accent as in Note 10

Line 464: “thirteenth-century” with hyphen

Line 472: The author does not mention Alonso Cano (1601-1667) and his painting of San Bernardo in the Prado.  The work falls outside the period under consideration but does show the durability of the compositional motif (the image in also conveniently the public domain).

Line 483: “María” with an accent here because it’s Spanish

Line 486: the preference now is for ‘Alfonso the Learned’, or just ‘Alfonso X’ (with or without ‘el Sabio’)

Line 492: “fol.” with period to show abbreviation

Line 503: “(Leyenda aurea fol. 35r; digital image 441 of 642)”: italics for title, correct presentation of folio number, not sure why inclusion of ‘digital image 441 of 642’

Line 508: perhaps ‘authority’ rather than ‘power’

Line 521: perhaps “the lactation vision granted to St. Bernard”

Line 533: lactatio in italics for Latin

Line 569: usually expressed as “had not yet made profession with the Dominicans”

Line 595: “rociado” is literally ‘bedewed’ so perhaps ‘moistened’

Line 610: perhaps “teaching” rather than “words”

Line 614: agudeza is perhaps better rendered as ‘insight[fulness]’ rather than the somewhat sterile ‘acuity’; a later generation will use agudeza as sharp wit.  The repeated use of learned forms of ‘mellifluous’ is a nod to Bernard’s fame as a Latin prose stylist and as a composer of hymn lyrics.

Line 628: Cantigas in italics

Line 636: perhaps ‘backdrop’ rather than ‘source’

The author may not wish to get any more scabrous than necessary, but the interrupted erotic interlude recounted in the Archpriest’s exemplum is also a parody of ejaculation.  The cuckolded husband gets an unwelcome jet of white body fluid squirted in his uncomprehending eyes, a calc for the usurped conjugal act another man has perform with the unfaithful wife.  Medieval readers suffered little of the squirming modern audiences might at that parallel.  They would have gotten it.  I tend to side with those who think the Archpriest’s blunt scorn for women – and wounded male entitlement – overshadows any pious intentions.

The author may be interested in this well-documented article which discusses the social practice of hiring lactating woman of proven moral quality to be wetnurses to the wealthy:

Bergmann, Emilie L. “Milking the Poor: Wet-Nursing and the Sexual Economy of Early Modern Spain”. Marriage and Sexuality in Medieval and Early Modern Iberia. Ed. Eukene Lacarra Lanz. NY & London: Routledge, 2001.